



# Technical note: GODESS – A profiling mooring in the Gotland Basin

Ralf D. Prien and Detlef E. Schulz-Bull

Leibniz Institute for Baltic Sea Research

*Correspondence to:* R. D. Prien
(ralf.prien@io-warnemuende.de)

**Abstract.** This note describes a profiling mooring with an interdisciplinary suite of sensors taking profiles between 180 m and 30 m depth. It consists of an underwater winch, moored below 180 m depth and a profiling instrumentation platform. In its described setup it can take about 200 profiles at pre–programmed times or intervals with one set of batteries. This allows studies over an extended period of time (e.g. two daily profiles over a time of three months). The Gotland Deep Environmental

5   Sampling Station (GODESS) in the Eastern Gotland Basin of the Baltic Sea is aimed at investigations of redoxcline dynamics. The described system can be readily adapted to other research foci by changing the profiling instrumentation platform and its payload.



## 1 Introduction

Assessing temporal dynamics in the sea on the order of hours to weeks is difficult to achieve with the classical oceanographic
tools; repeated CTD casts from the ship keep the ship bound close to position for extended times, classic moorings with
instruments at several depths are limited in their vertical resolution. This problem is exacerbated in dynamic systems such as
the redoxcline in the Baltic Sea with a succession of steep gradients in different properties. Profiling moorings have been used
for a long time and there are commercial systems available for full ocean depth (e.g. McLane moored profiler). Most of these
systems, however, use a buoyant body at the top end and a cable to the deep end as guidance for the profiling instrumentation
carrier. This makes such installations vulnerable for damage e.g. by trawled fishing nets. Here an underwater winch is used at
the lower end of the mooring and the profiling instrumentation carrier connected to the winch by a Kevlar line is ascending by
its own buoyancy through the water column before it is retracted by the underwater winch reeling the line back in.

### 1.1 Science application

The Gotland Basin, consisting of the Western Gotland Basin with the deepest position in the Baltic (459 m at Landsort deep)
and the Eastern Gotland Basin with a maximum depth of around 240 m, is the biggest deep basin of the Baltic. The Gotland
Basin is characterised by an upper layer of brackish water that is separated from the more haline waters in the deep by a strong,
permanent pycnocline. This pycnocline prevents vertical mixing and with it the transport of oxygen into the waters below.
Oxygen in the deeper layer is consumed by degradation of organic matter sinking from the upper water layers down through
the pycnocline (e.g. Nausch et al., 2008) and consequently a gradient in dissolved oxygen concentration builds up. Oxygen is
replenished mainly by major inflows of North Sea water that reach the central Baltic and replace the lower layer waters in the
deep basins (e.g. Feistel et al., 2003; Mohrholz et al., 2015). Repeated CTD casts over a period of time have shown, however,
that small scale variation of oxygen concentrations within the hypoxic down to the hypoxic/anoxic interface do occur (Jost
et al., 2007) also in the stagnant periods between major Baltic inflows.

To investigate these smaller scale dynamic signals in the redoxcline repeated measurements are needed to assess frequency
and amplitude of the small excursions of dissolved oxygen concentration and the consequences on the redox system. Repeated
CTD casts from the ship cannot be carried out over the longer periods of time that would be needed to be able to build statistics
and assess the dynamics under different seasonal and weather conditions. Consequently a profiling mooring was designed and
built that is able to autonomously take repeated profiles at pre–programmed times or intervals over a period of months (Prien
and Schulz-Bull, 2011). The mooring was placed in close proximity to a routine monitoring station in the Gotland Deep, thus
maximising the synergy from both stations; long term trends and information on important redox parameters in the standard
depths at the monitoring station and smaller scale variations from the profiling mooring nearby.



## 2 Moored system

A sketch of the profiling mooring is shown in figure 1. The mooring consists of the profiling instrumentation platform that
houses the instruments, the underwater winch, the recovery system and the bottom weight with ground line. This ground line
serves as deployment tool and emergency recovery fallback should the acoustic releaser fail.

Care has to be taken in the choice of materials for the mooring as all components are placed in anoxic waters containing
relatively high concentrations of $H_2S$ (up to around 80 $\mu$M (e.g. Nausch et al., 2008)). While this can pose a corrosion problem
even for high grade steels and restricts the choice of sensors on the instrumentation platform it mitigates bio–fouling issues
and decreases the risk of damage by fishing operations.

### 2.1 Profiling instrumentation platform PIP

The profiling instrumentation platform (PIP) accommodates the instrumentation payload that is shuttled through the water
column. It also provides the buoyancy necessary to keep the line to the underwater winch under tension and to provide the lift
for the instrument payload during the ascent through the water column. The net buoyancy of the instrumentation platform and
the payload instruments determines the ascent speed as well as descent speed and power demand of the underwater winch.

A custom build titanium frame, based on the standard design of Sea & Sun Technology's CTD frame, with additional fixtures
for the oxygen optode and syntactic foam sheets was used here for the first deployments. The standard frame from titanium
rods is 550 mm high with a 200 mm square cross section. This standard design was modified by pulling out one of the corner
rods of the square by another 200 mm, resulting in a kite shaped cross section with axes of 280 mm and 480 mm respectively.
A drawing of the profiling platform is shown in figure 2. Six syntactic foam sheets are attached to the frame. They provide the
buoyancy for the PIP as well as keeping it from rotation around the line axis as the end with the shorter buoyancy plates will
always will point towards the lateral current.

The syntactic foam has a density of 300 kg m$^{-3}$ resulting in a buoyancy force density of about 6870 N m$^{-3}$ in water. The six
sheets have a total volume of 23.25 litres and therefore a gross buoyancy of about 160 N. This is sufficient for the net buoyancy
of about 80 N that is needed to keep the line tension sensor in the underwater winch from shutting off the motor (see below).
The buoyancy was chosen to be sufficient for the line tension sensor with a small safety margin, but to minimise the buoyancy
and thus the ascent speed to accommodate also slower sensors on the PIP and to minimise the power requirement on the winch
for pulling the PIP back down into the parking position.

The PIP is attached to the line of the underwater winch using a stainless steel cable bridle. This keeps the platform upright
in the water column as a pre–deployment test in water showed. The shape of the platform was chosen to avoid it spinning in
the case of lateral flows.

### 2.2 Underwater winch

The underwater winch is an Automatic Elevator System Type 3 from Nichiyu Giken Kogyo Co., Ltd., Japan (see fig. 3). It's
maximum deployment depth is 300 m and it has a length of 360 m of a 2.7 mm diameter Kevlar line. It is 1.8 metres high with





a diameter of 1 metre and weighs 190 kg in air with battery packs installed. In water it is positively buoyant with a buoyancy of about 350 N.

At preprogrammed times or intervals the underwater winch unlocks the spool of Kevlar line that is pulled out by the buoyancy of the PIP, provided that the force pulling on the line is at least 80 N. Once the preset length is pulled out the winch stops for a set period of time before it starts reeling the Kevlar line back in. When the hook at the end of the line is latched in the parking position the winch control electronics is put in a wait state until the time of the next profile.

The minimum required pulling force is controlled by a line tension sensor located in the biggest of the winches buoyancy spheres. It stops the winch paying out line when the force gets smaller than the set value. This allows profiling up to the surface in different flow regimes or tidal states as well as profiling under ice up to the ice cover. Whenever the PIP is stopped in it's ascent the tension sensor switches off the motor and thus avoids loose lengths of line that could twist or entangle. Tension control also allows deploying the mooring with the PIP connected to the winch frame using corrosion links, making the handling during deployment easier. Profiling only starts when the corrosion links are broken and the PIP exerts a force on the line by it's buoyancy.

Power is supplied by Li–primary battery packs, one pack (60 Ah at 7.2 V) for the winch electronics and three packs of 60 Ah at 24 V each for the motor. The number of profiles that can be carried out with one set of batteries depends on the depth difference for each profile, the buoyancy, size and drag coefficient of the PIP and the flow field in the profiled water column. With the PIP described above 220 profiles of 156 m each or 34.3 km of profiled water column were achieved with one set of batteries. Additional battery packs can be added for increased endurance.

## 2.3 Recovery system

The recovery system for the profiling mooring consists of an acoustic releaser (KUM K/MT 572), a drum of 350 m 6 mm diameter Dynemaa recovery line (breaking strength of 27 kN) and buoyancy spheres. When the acoustic releaser is activated the safety pin of the drum is pulled and the Dynemaa line is spooled off the drum as the buoyancy of the PIP, the underwater winch and the buoyancy spheres attached to the releaser are pulling upward. Once PIP, winch and acoustic releaser are recovered from the surface the ground weight and ground line can be pulled up with the recovery line and all parts of the mooring are recovered.

## 2.4 Instrumentation

For the first deployments of the GODESS mooring the instrumentation payload consisted of a Sea & Sun Technology GmbH CTD90M with additional sensors: A Seapoint Sensors Inc. turbidity sensor, a Turner Designs Inc. Cyclops 7 fluorometer, an AMT GmbH oxidation–reduction–potential sensor and an AMT GmbH pH glass electrode sensor. Additionally a fast dissolved oxygen optode Rinko (JFE Advantech Co. Ltd) in a separate housing was installed on the PIP and connected to the CTD that provides the Rinko with power and logs the data.

The choice of sensors for the profiling mooring is limited by the anoxic conditions at the deep end of the profile, where the PIP is parked between profiles. The $H_2S$ in the anoxic waters can degrade sensor performance. $H_2S$–safe sensors also can suffer





these problems as manufacturers specify $H_2S$ resilience with the standard operation mode such as casts from a ship in mind; storing a sensor in $H_2S$–rich waters while not operating is not an application scenario considered routinely by manufacturers.

All sensor data are logged by the CTD90M at about 4 Hz. For those sensors fast enough, i.e. with a response time less than 0.25 s, the vertical resolution during the ascent of the PIP at 0.36 m s$^{-1}$ (typical for the first deployments) is about 9 cm. The CTD's internal memory (64 megabytes) is sufficient for at least 80 hours of logging at this rate and therefore no limitation for the system.

Power for all sensors is provided by 8 size C alkaline batteries in the CTD. Power consumption for the whole sensor suite including the Rinko oxygen optode is 0.92 W during measurement. The battery power is sufficient for 44 hours of measurement (220 profiles with recording intervals of 12 minute length each), the power consumption during the intervals between profiles is negligible. The battery voltage drop during the second deployment (figure 4) over time suggests that a total recording time of about 50 hours would have been possible before the supply voltage drops below 9.5 V, still 0.5 V higher than the threshold for the CTD electronics to switch itself off.

The CTD and oxygen sensor were installed in the PIP side by side with the built–in sensors of the CTD and the sensing surface of the oxygen sensor pointing upwards.

## 2.5 Operation

Recording of the sensor signals in the CTD has to be synchronised with the operation of the underwater winch. As the two units have no electrical or other communications link they are synchronised by their real time clocks (RTC). Neither the underwater winch nor the CTD features a high precision RTC but the first deployments allowed to determine the drift between the clocks to be in the order of 1.3 s per day (with the CTD clock lagging behind the winch clock). This drift, however, is likely dependent on the temperature during the deployment. As the temperature variation in 180 m depth is very small it is safe to assume the same drift rates for future deployments of the same instruments. When the CTD is programmed to start two minutes ahead of the winch starting to release line all profiles over a deployment of three months should be captured in full at a recording interval length that is two minutes longer than the ascent time of the PIP.

The time between profiles depends on the measurement task; in the three deployments undertaken so far it has been one hour for a short 17 hour test deployment, four hours in the second deployment and eight hours for the third. The maximum endurance for the three deployments at 220 profiles would have been 9 days, 36 days and 73 days, respectively.

## 3 Results

### 3.1 First deployment

A first test deployment at the GODESS position in the Gotland Deep was carried out in May 2010. The mooring layout was modified in that the recovery system was not installed; the end of the ground line was attached to a second ground weight with a further line with buoyancy spheres and a surface marker buoy attached. Recovery was carried out by latching on to the surface



marker buoy and bringing in the line, second ground weight, ground line, winch and PIP. During the 17 hour deployment the winch was programmed to carry out hourly profiles from about 150 m to 3 m depth. For the first deployment the CTD was left running during the whole of the deployment, data for the ascents as well as the descents and parking phases exist. During normal deployments the CTD starts measurements shortly before the ascent commences and stops shortly after the ascent has

finished to conserve energy and storage. The data of the periods between profiles gained in this first deployment show that the PIP stays at depth without much movement (the peak to peak variation of pressure is about 0.07 dbar) so it can be assumed that lateral currents were very low.

The ascent is highly reproducible in speed, only the first profiles show some deviation at the beginning, being a little slower at the start of the ascent. The ascent speed is about $0.36\,\mathrm{m\,s^{-1}}$ on average, starting at about $0.37\,\mathrm{m\,s^{-1}}$ at 150 m and slowing

down to about $0.33\,\mathrm{m\,s^{-1}}$ at 3 m. The reason for the slow down during ascent is thought to be the change of leverage on the underwater winches drum exerted by the Kevlar line as it is spooled off by the pull of the PIP's buoyancy. At the beginning the layer of spooled line is thick (i.e. leverage high), thinning as the line is paid out. The buoyancy change of the PIP due to density changes in the surrounding water does work in the same direction but the density change from about $1010.1\,\mathrm{kg\,m^{-3}}$ at 150 m depth to about $1005.8\,\mathrm{kg\,m^{-3}}$ at 3 m depth is too small (about 0.4%) to explain the differences in ascent speed. On the

way down the speed is fairly constant at about $0.16\,\mathrm{m\,s^{-1}}$. For the routine deployments the data will be taken on the upcast, for sensors with slower response times it might be advantageous to change the PIP so that the measurements are taken on the descent. Figure 5 shows the temporal differentials of measured pressure during the first deployment.

## 3.2  Second deployment

The second deployment was the first test of the complete system, i.e. including the recovery system. Figure 6 shows the main

components of the profiling mooring on deck before deployment on 3 July 2010. Recovery commenced on 5 August 2010 (33 days after deployment). The system recorded 198 profiles from about 185 m depth to about 40 m depth, one profile every four hours. The performance during ascent was comparable to the first deployment, showing the same slow down of the ascent at shallower depths and also the slower start of the ascent for the first few profiles. The average ascent speed was again about $0.36\,\mathrm{m\,s^{-1}}$.

All sensors worked well for the whole deployment, only the data from the pH sensor show unexpected variation in the deep

that can be attributed to the sensor operation (see discussion below). The inertial period of the Gotland basin (about 14 hours) can be clearly seen in most sensor signals (T (see fig. 7), S, dissolved oxygen (see fig. 8), ORP, turbidity). The minimum depth recorded by the CTD on the PIP does vary very little, a clear sign that the currents were low. Around 17 July 2010 the minimum depth reached by the PIP is about two metres deeper than usual, at the same time the redoxcline is shifted downwards by the same amount.

Figure 7 shows the temperature plotted colour coded over deployment time and pressure. At the minimum depth of the PIP, around the 40 dbar level, low temperatures of around 2.5 °C are measured; this cool water between 40 and 60 dbar is the winter water, cooled during the previous winter it sank down to the pycnocline at around 70 dbar. The warmest water in the profiles can be found around the 120 dbar level at about 7.2 °C. Prominent in this figure are the often short lived colder periods within



this warm level between around 100 and 140 dbar. These can be present in one profile and absent in the next one, taken four hours later. It is this dynamic change that makes it necessary to take repeated profiles over a longer time frame to establish the statistics of the fluctuations.

In figure 8 the dissolved oxygen concentration is plotted colour coded over deployment time and pressure. All concentrations above 90 mmol m$^{-3}$ are shown in red to spread the colour scale, emphasizing the concentration changes in the hypoxic regions. When comparing the structures visible in the hypoxic regions with the structures seen in fig. 7 it becomes clear that the colder intrusions are accompanied by higher dissolved oxygen concentrations. The most prominent feature is the patch of higher dissolved oxygen concentrations starting at around 120 hours into the deployment (8 July 2010) and lasting for about 80 hours. Judging by the structure of the feature at a depth level of about 85 dbar this event is certainly dominated by lateral movement through the profiling position. On 24 July 2010 a shorter lived event reaches down some 20 dbar into the anoxic zone.

### 3.3 Third deployment

The third deployment was carried out on 16 November 2010 and the system recorded 170 profiles from about 183 m depth to about 37 m depth, one profile every eight hours. When recovery took place on 7 February 2011 the PIP was found drifting on the sea surface. The winch had been programmed to take profiles until the 11 February 2011 (270 profiles) well aware that this most likely would be above the battery capacity. The winch log files revealed that the winch was operating until the 28 January 2011 (221 profiles) when the batteries were depleted. This number of profiles is in good agreement with the estimates based on the previous deployment and battery capacities.

When downloading the data from the CTD's memory it showed that the CTD stopped recording profiles already on 11 January 2011, well before the anticipated end of battery capacity. Inspection of the battery voltage record shows two events where the battery voltage seems to suddenly decrease (see fig. 9). At the same times the pH profiles showed a different shape than preceding or following ones. Inspection of the CTD at the manufacturers revealed the most likely cause of these events. At both occasions (and probably after the final profile recorded) the CTD did not disable power to the sensors during the almost eight hours between successive profiles; after the following profile, however, the CTD did switch off the sensors again. It can be assumed that the same happened again after the last recorded profile and during the eight hours the battery voltage dropped below the CTD's threshold of 9 V so that the CTD stopped recording after that time.

Looking at the different shape of the pH profiles immediately before and after the external sensors had been powered during the time between profiles (see fig. 10) it becomes clear that the profile taken after the sensor was powered continuously for eight hours looks more reasonable; the measured pH remains constant in the anoxic water below approx. 125 dbar as would be expected, the slow decrease of pH in the earlier profile is assumed to be an artefact due to the storage of the pH electrode sensor in waters containing H$_2$S at concentrations around 80 $\mu$Ml$^{-1}$. While the pH electrode was specified as H$_2$S–safe, the manufacturer had not anticipated the mode of operation employed for the profiling mooring; the H$_2$S resilience was stated for the more classical scenario of the powered pH electrode being introduced in the medium containing H$_2$S only for measurement and retrieved from the medium before powering off.





Assuming fairly constant parameters including $H_2S$ at the parking depth the same running-in characteristics can be expected after powering the pH electrode. Thus the pH profiles can be corrected by fitting a third order polynomial to the difference

between the two successive profiles and adding the resulting correction to the profiles taken after the power had been switched off between profiles. The interpretation of the corrected pH data, however, should be treated with great care. For future deployments the pH electrode will be powered continuously. As the pH electrode is a low power sensor this will decrease the potential recording time only by a negligible amount. The voltage drops experienced during the periods between profiles when the power didn't switch off during this deployment were mainly due to the oxygen sensor that has a power consumption of

0.6 W.

## 4   Conclusions and discussion

The GODESS profiling mooring has been deployed successfully in the Gotland Deep of the Baltic Sea taking 385 profiles over a deployment time of 89 days in total. The profiling mooring allowed for the first time the registration of the dynamics of oxygen and other redox–relevant parameters with high vertical and temporal resolution over an extended period of time in

summer and winter situations. The datasets allow detailed studies of the dynamics of the redoxcline and are a valuable resource for biogeochemical redoxcline investigations. Routinely employed the data from the profiling mooring will complement the ship based sampling and analysis and provide an environmental context for ship campaigns.

In the first deployments the underwater winch and the PIP with its instrumentation worked reliably. The first deployments did exhibit, however, some handling problems on deployment and recovery that could jeopardize deployment success. Thus

a re–design of the PIP will be carried out to improve handling of the system and also to incorporate additional sensors. A UV/vis absorption spectrometer will allow high resolution measurements of e.g. nitrate and $HS^-$ Prien et al. (2009). Wet chemical analysers for Mn(II) and Fe(II) (Meyer et al., 2011; Prien et al., 2006) will also in future be added on some shorter deployments. The addition of a current meter is a further planned addition that will allow to determine the direction of origin of lateral intrusions.

With the current PIP and sensor suite the system has been proven as a useful tool for high temporal and vertical resolution investigations of the redoxcline. It can assess small scale dynamics in all weather conditions, i.e. also in situations when work

from a ship is not possible. Taking profiles rather than measuring at multiple discrete depth levels has the advantage that recorded parameters can easily be referred to densities instead of pressure or depth. Profiling moorings like GODESS can also satisfy the requirements regarding temporal resolution that have been identified as necessary for a future Baltic monitoring system (Karlson et al., 2009).

The system is versatile as the PIP can be easily replaced by another one with a different sensor suite to target other science

applications. The GODESS mooring also is a highly attractive test bed for new in situ sensors as the Gotland Basin redoxcline exhibits strong gradients in a number of potential target parameters as well as variations of parameters potentially causing cross–sensitivities.



*Acknowledgements.* The GODESS mooring was one of the field observatories of the EU project HYPOX (EC grant 226213). The hardware was funded through the ZIP programme of the state of Mecklenburg–Vorpommern. The authors would like to thank Uwe Hehl for help with
15 the mooring design and all aspects of deployment and recovery. Siggi Krüger was very helpful in many discussions on the system. The crews of the research vessels Prof. Albrecht Penck, Alkor, Heincke and Maria S. Merian we thank for excellent service and seamanship.





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





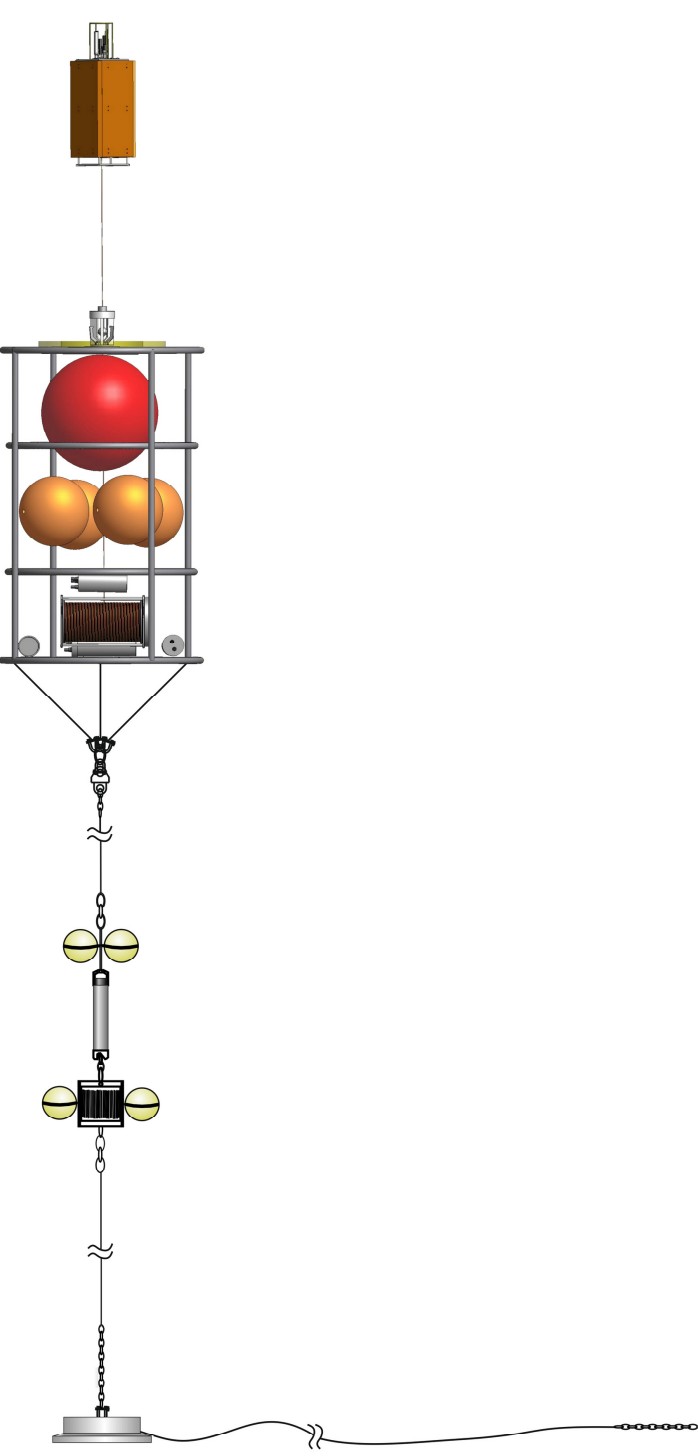

**Figure 1.** Sketch of the mooring with profiling instrumentation platform (PIP), underwater winch, acoustic releaser with recovery line and bottom weight with ground line (from top to bottom).



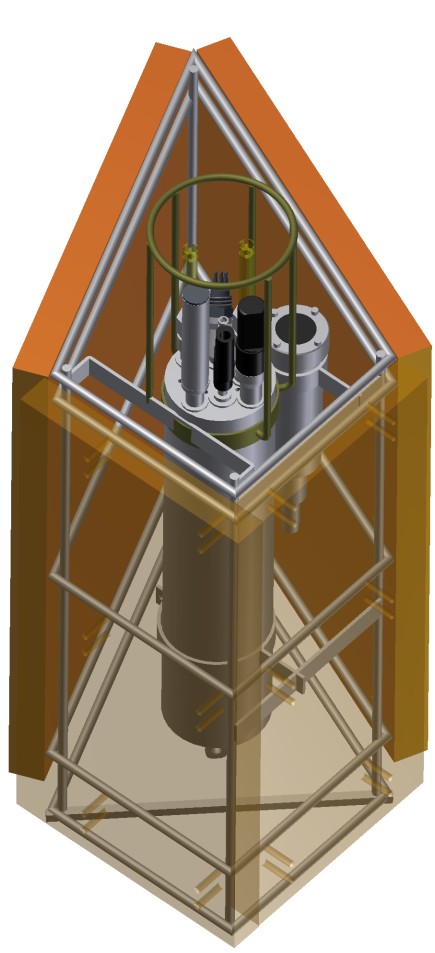

**Figure 2.** CAD drawing of the PIP frame with installed CTD and oxygen optode. Front sheets of syntactic foam are shown as transparent.





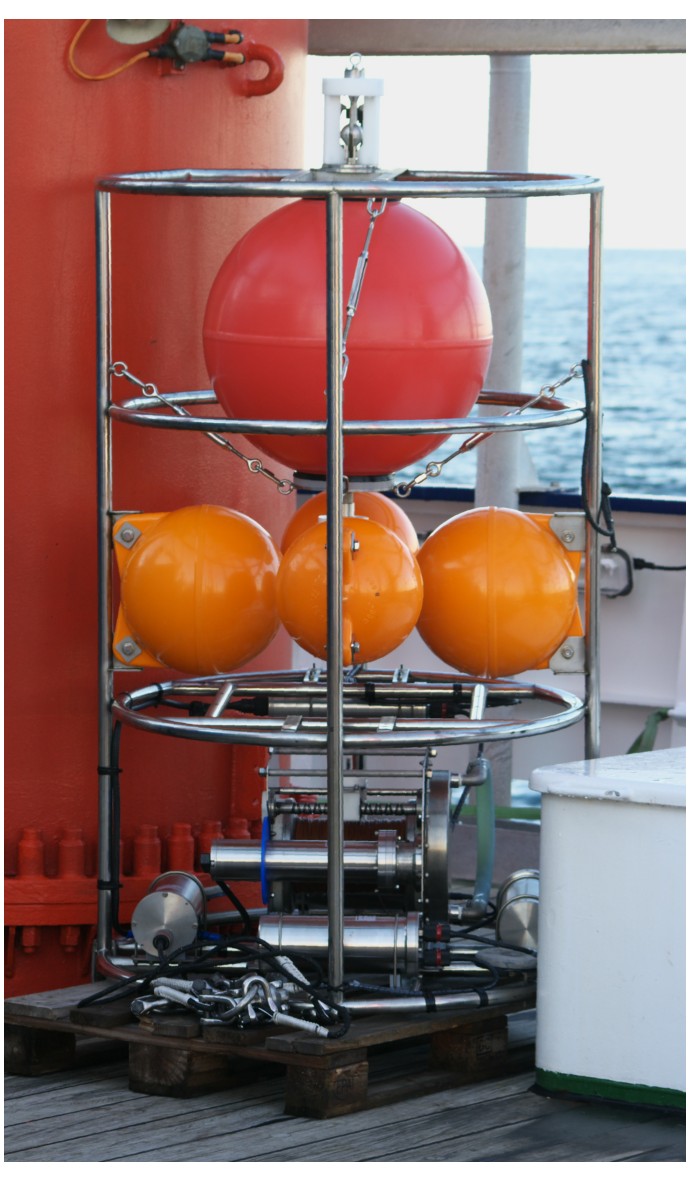

**Figure 3.** Underwater winch on deck after recovery from the second deployment in August 2010.





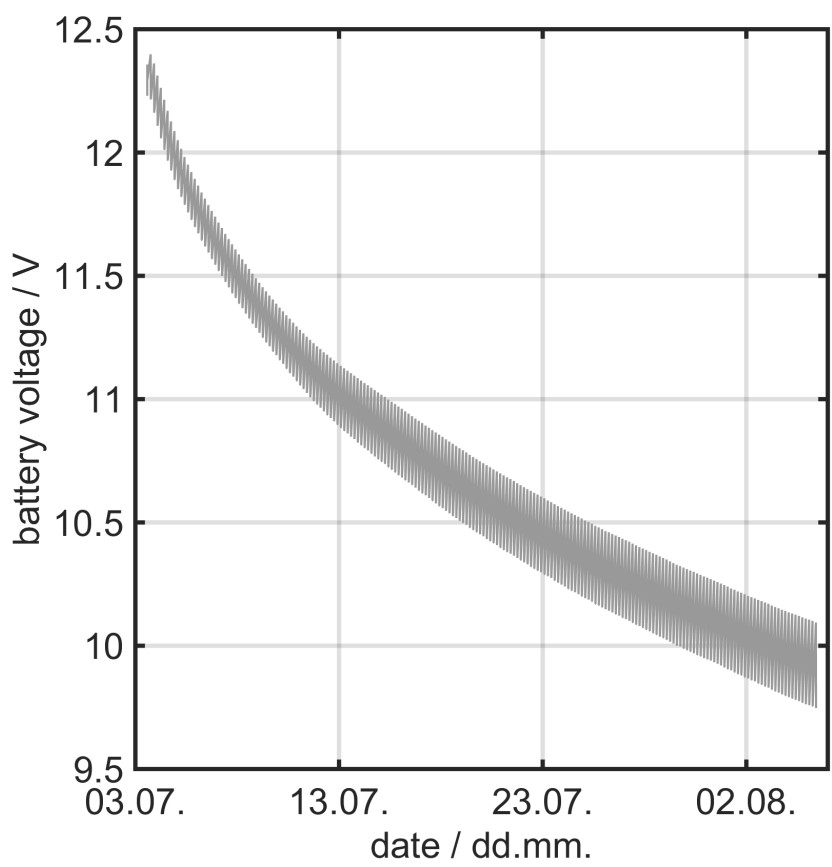

**Figure 4.** Battery voltage over the time of the second deployment in July/August 2010 (grey line). The voltage drops during profiling and recovers during the 4 hour sleep period.





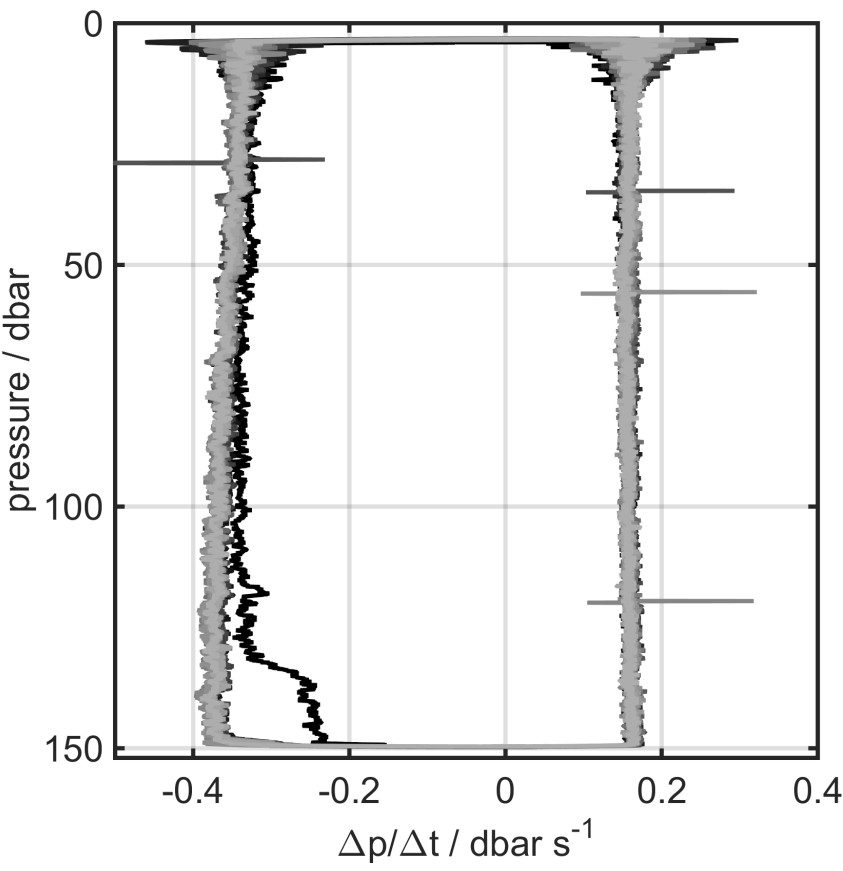

**Figure 5.** Ascent speed (in dbar s$^{-1}$) for the 17 profiles of the first deployment. Profiles are colour coded, changing gradually from the first profile in black to the last one in light grey.



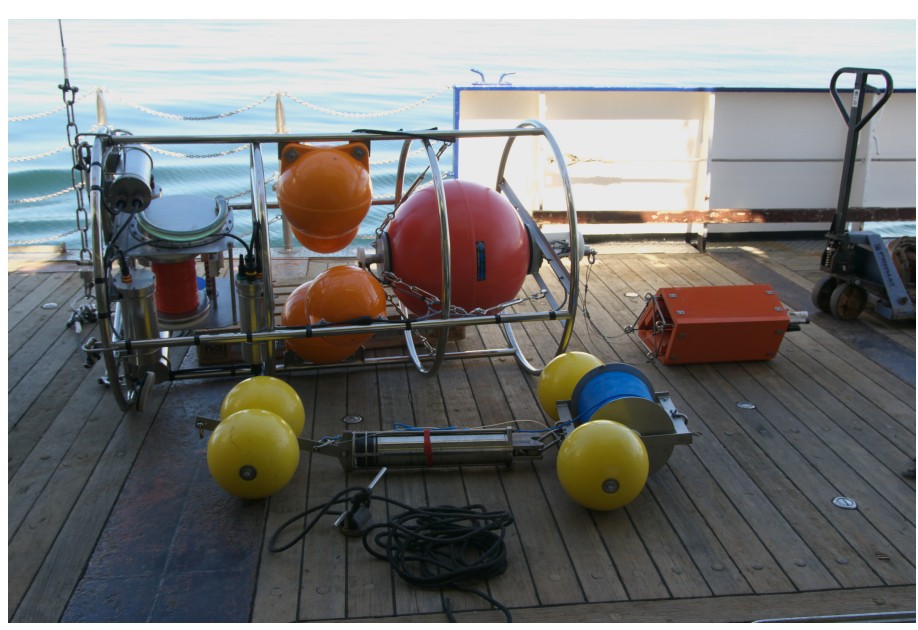

**Figure 6.** Mooring components laid out on deck before the second deployment in July 2010. Recovery system with acoustic releaser in the front, underwater winch back left and profiling platform with instruments back right.





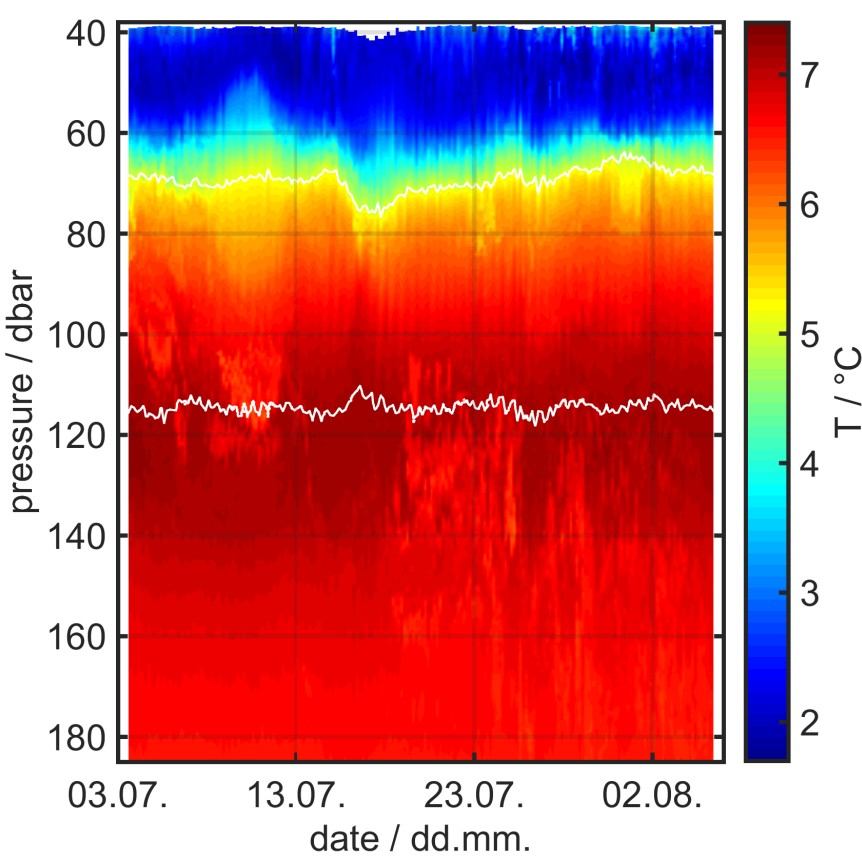

**Figure 7.** Plot of temperature (colour coded) over deployment date and pressure for deployment 2, 3. July - 5. August 2010. The white lines at around 70 dbar and around 115 dbar denote the isopycnals of 1007.6 $kg\,m^{-3}$ and 1009.5 $kg\,m^{-3}$ respectively, indicating the division between oxic and hypoxic (upper line) and between hypoxic and anoxic waters (lower line).





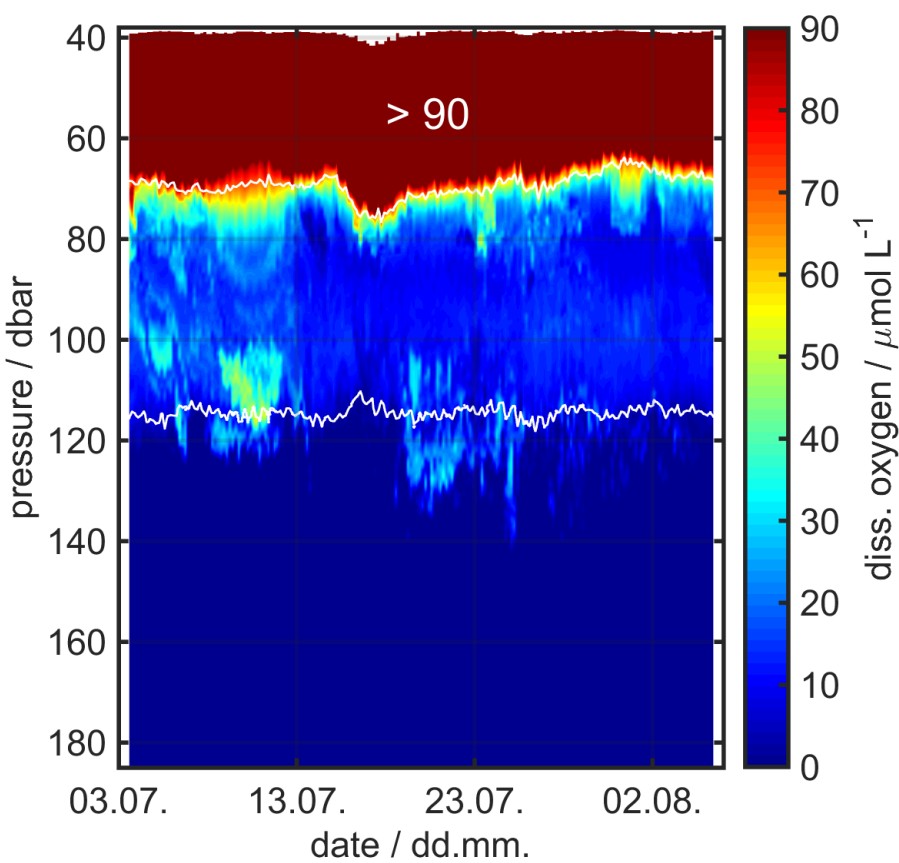

**Figure 8.** Plot of dissolved oxygen concentration (colour coded) over deployment date and pressure. All concentrations above 90 $\mu\mathrm{mol\,m}^{-3}$ (the threshold between hypoxic and oxic waters) plotted in the same colour to emphasize dynamics in the hypoxic waters. The white lines at around 70 dbar and around 115 dbar denote the isopycnals of 1007.6 $\mathrm{kg\,m}^{-3}$ and 1009.5 $\mathrm{kg\,m}^{-3}$ respectively, indicating the division between oxic and hypoxic (upper line) and between hypoxic and anoxic waters (lower line).





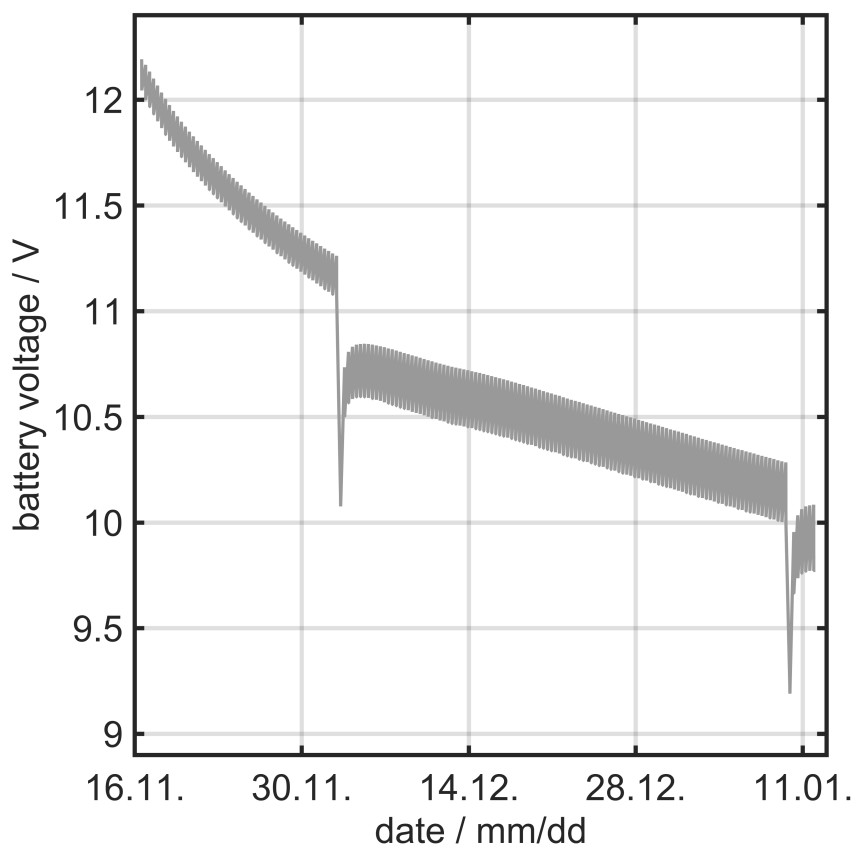

**Figure 9.** Battery voltage over the time of the third deployment from November 2010 to January 2011. The voltage usually drops during profiling and recovers during the 8 hour sleep period. At December 3[rd] and January 9[th] seemingly sudden drops in battery voltage occurred.





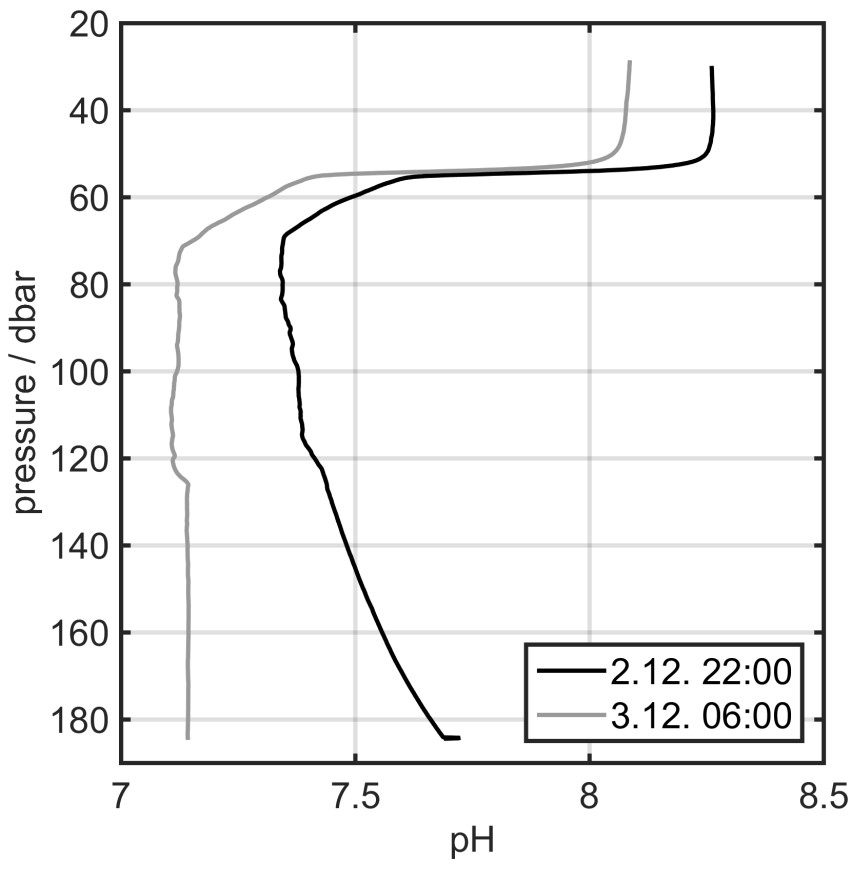

**Figure 10.** Two successive profiles of pH, taken before the CTD failed to switch off power to the external sensors (December 2[nd] 22:00 h, black) and eight hours later after sensors had been powered for eight hours (December 3[rd] 06:00 h, grey).