# Peer review of "Technical note: GODESS – A profiling mooring in the Gotland Basin"

_Ocean Science, 2016_

## Referee Comment (RC1) · J. Toole (Referee) · 14 Mar 2016

Review of "Technical note: GODESS – A profiling mooring in the Gotland Basin" by Ralf D. Prien and Detlef E. Schulz-Bull

The authors present a straightforward description of their moored winch system designed to autonomously collect multiple profiles of water properties in the Baltic Sea over time periods of several weeks to months along with the results from 3 test deployments. The GODESS system represents an integration of several commercially-available components plus some adaptations for the specific measurement site. The paper's introduction makes note of previous "wire-following" moored profiling instruments, but fails to review past (and current) moored winch systems. There are in fact many of these (both commercially-available and "home-built"). It seems to me that the

principal difference between Prien and Schulz-Bull's system and these other moored winch systems (that I know about) are the adaptations for the unique water properties of the deep Baltic Sea (high H2S concentrations in particular). I would have welcomed in this paper a review of the state of moored winch systems that highlights the novel features of the GODESS mooring. A major shortcoming of the Prien/Schulz-Bull system in my opinion is lack of communication between the winch and profiling instrumentation package (PIP). Apart from the synchronization issues (addressed by the authors by relying on clock stability in the two components), at present, all of the science data collected by their device are stored in the PIP. Should the tether break (as could easily happen with the PIP left floating at the surface after the winch battery is exhausted (as occurred in their third test), all of the information would be lost. The moored winch systems I am familiar with address this communication link using an inductive modem (operational when the profiling unit is retracted against the winch module). Perhaps the authors will look into adding this capability in future? Also, couldn't the winch controller monitor battery level and terminate operations with the PIP (somehow) locked in its retracted position before the battery is exhausted fully? My other more general concern about this technology is its broader applicability. The Baltic experiences rather weak currents and small surface waves (relative to many coastal ocean sites). Currents can prevent the low-buoyancy PIP from reaching near the surface. Wave action can produce repeated periods of slack tether tension that can cause line snarls at the winch. The ocean is a difficult environment to make measurements. No instrument system is perfect. The GODESS mooring described in this paper shows promise for sustained observations in the Baltic. I believe this paper could be strengthened by a more thorough discussion of its present shortcomings and sketching future development work to address them.

---

## Author Comment (AC1) · 21 Apr 2016

Thank you for reviewing our manuscript "Technical note: GODESS - A profiling mooring in the Gotland Basin". We are sure that modifications to the manuscript, triggered by your comments, will improve the manuscript.

Yes, a review of the state of moored winch systems would be a good thing. Such a review would be a separate manuscript, though, as a Technical note in Ocean Science is supposed to be "short (a few pages only)" as stated in the description of manuscript types. Ideally it would also need a wider pool of authors to draw in experience with the different profiling mooring types and concepts. In a review the technical details of our GODESS mooring would also be too detailed.

A link between the PIP and the winch using inductive modems is certainly feasible

(and has been realized for this type of underwater winch by another group successfully (B. Fiedler, pers. comm.)). This can be achieved either by replacing the Kevlar line by a cable or by adding inductive couplers on the latch for the hook of the winch and the lower part of the PIP (as you suggest in the comment). In both cases the data logging would be doubled up, as the Sea & Sun CTD already contains a data logger. The added safety for data comes with the increased complexity and increased power demands (albeit a very moderate increase compared to the total energy installed on the winch already).

The risk of loss of data could be decreased also by adding a satellite beacon on the PIP, that would be sending position data once the PIP is drifting on the surface. As these beacons are self-contained with own battery supply it wouldn't increase the complexity of the system. Admittedly this solution is more feasible in the Baltic Sea, where a drifting PIP cannot get it away and it would be much easier to find a research vessel nearby that could pick it up.

The suggested monitoring of battery level could most probably be realized. The information on the controller electronics used in the winch is sparse, though, and it would necessitate a complete redesign of the controller electronics. Since we now have the experience of how much energy is taken out of the batteries (and we usually measure the energy left in the primary batteries after a deployment) the risk of running out of energy is well reduced to a failing battery or exceptionally high currents that demand an increased motor current for reeling the PIP back in.

We will endeavour to add some of the points you raised in the discussion to make the manuscript useful for a wider audience.

---

## Referee Comment (RC2) · R. Pinkel (Referee) · 11 May 2016

At your request I have reviewed the article *GODESS – A Profiling Mooring in the Gotland Basin* by Prien and Schulz-Bull, et al.  In general this is an excellent well-written article of significant interest.  As a physical oceanographer, I am interested in hearing more about the expected performance of the system in stronger currents.  Is there some estimate of the drag of the profiler that would enable calculation of how much the profiler would lay over while profiling in a current?  If so, the proximity to the sea surface and the energy required to pull the float downward / inward could be calculated vs. current speed.  These numbers would be of great interest to those developing profiling systems.

Additionally the authors' experience with the Sea and Sun Technology CTD is of general interest. Many of us are hoping to find a CTD that consumes less power than a "pumped" SeaBird system. Could the authors show some successive profiles of salinity from the CTD and perhaps a map similar to Figure 7 for salinity?  This would augment the very valuable discussion of the pH sensor.

I appreciate John Toole's comments regarding developing a communications and feedback system between the sensor package and the winch. The system would become far more capable, but far more expensive / complex, as well.

The only technical error I noted was that the white line supposedly drawn at 70 m on Figure 7 is not visible on my graph.

With this issue corrected and perhaps the suggested augmentation included, the paper is ready for publication.

---

## Author Response (AR1)

**Reply to interactive comments**

R.D. Prien, D.E. Schulz-Bull

June 7, 2016

**Reviewer comments 1**

1. *It seems to me that the principal difference between Prien and Schulz-Bulls system and these other moored winch systems (that I know about) are the adaptations for the unique water properties of the deep Baltic Sea (high $H_2S$ concentrations in particular).*

   Yes, this is a fair reflection.

2. *I would have welcomed in this paper a review of the state of moored winch systems that highlights the novel features of the GODESS mooring.*

   Yes, a review of the state of moored winch systems would be a good thing. Such a review would be a separate manuscript, though, as a Technical note in Ocean Science is supposed to be "short (a few pages only)", as stated in the description of manuscript types. Ideally it would also need a wider pool of authors to draw in experience with the different profiling mooring types and concepts. In a review the technical details of our GODESS mooring would also be too detailed.

3. *A major shortcoming of the Prien/Schulz-Bull system in my opinion is lack of communication between the winch and profiling instrumentation package (PIP). Apart from the synchronization issues (addressed by the authors by relying on clock stability in the two components), at present, all of the science data collected by their device are stored in the PIP. Should the tether break (as could easily happen with the PIP left floating at the surface after the winch battery is exhausted (as occurred in their third test), all of the information would be lost. The moored winch systems I am familiar with address this communication link using an inductive modem (operational when the profiling unit is retracted against the winch module). Perhaps the authors will look into adding this capability in future?*

A link between the PIP and the winch using inductive modems is certainly feasible (and has been realized for this type of underwater winch by another group successfully (B. Fiedler, pers. comm.)). This can be achieved either by replacing the Kevlar line by a cable or by adding inductive couplers on the latch for the hook of the winch and the lower part of the PIP (as you suggest in the comment). In both cases the data logging would be doubled up, as the Sea & Sun CTD already contains a data logger. The added safety for data comes with the increased complexity and increased power demands (albeit a very moderate increase compared to the total energy installed on the winch already). The danger of loosing the PIP due to a breaking tether while floating on the surface is reduced by two factors: Firstly, from previous deployments we now have a good estimate of the energy needed for a profile (and regularly check how much energy was left in the batteries after the deployment). Secondly, we will use a reduced length tether so that in the case of depleted batteries the PIP will not float at the surface. (This second factor would be void in cases where the surface layer properties are of interest and/or where the data or parts of the data would be transmitted via radio or satellite link.) The risk of loss of data could be decreased also by adding a satellite beacon on the PIP, that would be sending position data once the PIP is drifting on the surface. As these beacons are self-contained with own battery supply it wouldnt increase the complexity of the system. Admittedly this solution is more feasible in the Baltic Sea, where a drifting PIP cannot get it away and it would be much easier to find a research vessel nearby that could pick it up.

We have added some of these thoughts in the discussion.

4. *Also, couldn't the winch controller monitor battery level and terminate operations with the PIP (somehow) locked in its retracted position before the battery is exhausted fully?*

The suggested monitoring of battery level could most probably be realized. The information on the controller electronics used in the winch is sparse, though, and it would necessitate a complete redesign of the controller electronics. Since we now have the experience of how much energy is taken out of the batteries (and we usually measure the energy left in the primary batteries after a deployment) the risk of running out of energy is well reduced to a failing battery or exceptionally high water currents that demand an increased motor current for reeling the PIP back in.

5. *My other more general concern about this technology is its broader applicability. The Baltic experiences rather weak currents and small surface waves (relative to many coastal*

> *ocean sites). Currents can prevent the low-buoyancy PIP from reaching near the surface. Wave action can produce repeated periods of slack tether tension that can cause line snarls at the winch.*

The applicability for high current and high wave regions certainly is limited or the operation of such a system in such conditions only possible running a greater risk of loss. However, we designed the system to operate in our region of interest and without the intention to profile up to the surface. That stated, the Alfred–Wegener–Institute did deploy moorings with the NGK winch system at the top end with success in the Arctic ocean over a year with their PIP proceeding to the surface regularly (see `https://www.awi.de/en/science/climate-sciences/physical-oceanography/instruments/underwater-winch/performance-report.html` and/or `http://epic.awi.de/28932/1/Bud2009a.pdf` (Chapter 3b) accessed on 31. May 2016, Gereon Th. Budeus (2009): Autonomous daily CTD profiles between 3,700 meters and the ocean surface. Sea Technology, 45-48. )

6.     *The GODESS mooring described in this paper shows promise for sustained observations in the Baltic. I believe this paper could be strengthened by a more thorough discussion of its present shortcomings and sketching future development work to address them.*

The We have added in the discussion some thoughts and potential changes to the system that may extend the applicability of a similar profiling mooring to less benign areas.

**Reviewer comments 2**

1.     *As a physical oceanographer, I am interested in hearing more about the expected performance of the system in stronger currents. Is there some estimate of the drag of the profiler that would enable calculation of how much the profiler would lay over while profiling in a current? If so, the proximity to the sea surface and the energy required to pull the float downward / inward could be calculated vs. current speed. These numbers would be of great interest to those developing profiling systems.*

This is a difficult one to answer for us as we are using the profiler only in low current regimes (about 0.3 m/s to 0.4 m/s max.) and have no experience in higher currents. Meanwhile we have carried out profiles with a new PIP design (PIP2) carrying additional instruments including a current profiler that incorporates an inclination sensor.

This PIP2 does show a pitch of +- 5 degrees and a roll of around 3 degrees. Dedicated calculations of profiler drag (and a wind tunnel test of a model) we have only carried out for a new design of PIP (PIP3) using a Wortmann profile shape. For our deployment area, though, the PIP2 design is certainly sufficient and the inclination data from this PIP2 reinforce our belief that the PIP1 design was sufficiently suited to the environment as well.

2. *Additionally the authors experience with the Sea and Sun Technology CTD is of general interest. Many of us are hoping to find a CTD that consumes less power than a "pumped" SeaBird system. Could the authors show some successive profiles of salinity from the CTD and perhaps a map similar to Figure 7 for salinity? This would augment the very valuable discussion of the pH sensor.*

We have added a figure for salinity but are well aware that this only gives a qualitative look at the performance of the Sea and Sun Technology CTD M90. A more quantitative analysis could perhaps best be carried out in a separate technical note. We have also added as an additional resource an animation in a movie file, `oxy_T_S_sig_movie.avi`, showing dissolved oxygen, temperature and salinity profiles for our second deployment plotted as a function of potential density $\sigma_T$. On the salinity panel we overlaid the differences to the previous profile.

3. *I appreciate John Toole's comments regarding developing a communications and feedback system between the sensor package and the winch. The system would become far more capable, but far more expensive / complex, as well.*

We do agree with this statement, it has been shown that this system does work for our region of interest. The addition of a satellite beacon will increase safety in case of a loss of PIP. Apart from that addition we rather concentrate on expanding the range of sensors carried on the PIP.

4. *The only technical error I noted was that the white line supposedly drawn at 70 m on Figure 7 is not visible on my graph.*

We have changed the colour of the upper isopycnal in the graphs to black to make it stand out better.

[revised manuscript text omitted]